# Fabrication of Poly(Vinylidene Fluoride)/Graphene Nano-Composite Micro-Parts with Increased β-Phase and Enhanced Toughness via Micro-Injection Molding

**DOI:** 10.3390/polym13193292

**Published:** 2021-09-27

**Authors:** Wu Guo, Zhaogang Liu, Yan Zhu, Li Li

**Affiliations:** State Key Laboratory of Polymer Materials Engineering, Polymer Research Institute of Sichuan University, Chengdu 610065, China; scu2020223095177@163.com (W.G.); 18561481381@163.com (Z.L.); anyewo@163.com (Y.Z.)

**Keywords:** poly(vinylidene fluoride), graphene, nano-composite, micro-injection molding

## Abstract

Based on poly(vinylidene fluoride)/graphene (PVDF/GP) nano-composite powder, with high β-phase content (>90%), prepared on our self-designed pan-mill mechanochemical reactor, the micro-injection molding of PVDF/GP composite was successfully realized and micro-parts with good replication and dimensional stability were achieved. The filling behaviors and the structure evolution of the composite during the extremely narrow channel of the micro-injection molding were systematically studied. In contrast to conventional injection molding, the extremely high injection speed and small cavity of micro-injection molding produced a high shear force and cooling rate, leading to the obvious “skin-core” structure of the micro-parts and the orientation of both PVDF and GP in the shear layer, thus, endowing the micro-parts with a higher melting point and crystallinity and also inducing the transformation of more α-phase PVDF to β-phase. At the injection speed of 500 mm/s, the β-phase PVDF in the micro-part was 78%, almost two times of that in the macro-part, which was beneficial to improve the dielectric properties. The micro-part had the higher tensile strength (57.6 MPa) and elongation at break (53.6%) than those of the macro-part, due to its increased crystallinity and β-phase content.

## 1. Introduction

With the mushrooming growth of the electronic industry, the miniaturization and light weight of electronic products have become research hotspots [1], thus, producing cutting-edge technology in the 21st century, i.e., micro-electromechanical systems (MEMS) technology, as well as the rapid development of devices toward miniaturization and functionalization [2]. Due to the low melting point, low density, accurate replication, low cost and good processability, polymer materials have been widely used in the field of MEMS, and polymer micro-devices have become one of the fastest growing and most promising products [3]. Injection molding, hot press molding and melt extrusion can all process polymer micro-devices, among which, micro-injection molding has attracted more attention for its advantages of good product quality, short molding cycle, low processing cost and easy industrialization [4].

At present, the corresponding micro-products prepared via micro-injection molding have been widely used in the fields of micro-optics, micro-actuators, biomedicine, microelectronics and aerospace. However, because of its different processing conditions from conventional injection molding, i.e., extremely small runner and cavity dimension, short filling time (only 10 ms), fast cooling speed, high injection speed (up to 1200 mm/s) and high shear rate (up to 10^6^/s) [5], polymer materials used for micro-injection molding are limited to materials with relatively good fluidity. For example, polymethyl methacrylate (PMMA) [6], polyoxymethylene (POM) [7], polyethylene (PE) [8], polypropylene (PP) [9], poly(lactic acid) [10], etc. It is difficult to meet the functional requirements of micro devices for mechanical, optical, electrical, magnetic or biomedical purposes. Therefore, polymer based micro-nano composites with high performance, multi-functionality and suitable for micro-injection molding are urgently needed [11]. In this situation, the micro-injection molding of carbon nanotubes (CNTs) composites, including polycarbonate/CNTs, polylactide/CNTs, liquid crystal polymer/CNTs, etc., for functional and structural applications have been studied [12,13,14,15,16].

Poly(vinylidene fluoride) (PVDF) has strong piezoelectric, dielectric and ferroelectric properties, so is usually used to prepare sensors, brakes and energy storage capacitors, and is expected to be a new type of polymer-based material for micro-injection molding by compounding with other functional fillers [17]. For example, the incorporation of halloysite nanotubes (HNTs) into a PVDF matrix improved the electroactive β- and γ-phases of PVDF via the nucleation effect of HNTs [18]; the sandwich structure obtained via the addition of 2D MXene and 2D hexagonal BN nanosheets endowed PVDF with high dielectric and breakdown properties [19]; the compositing with piezoelectric ceramics (PZT, BaTiO_3_, ZnO) or conductive particles (metals, inorganic salts, or carbon material) improved the piezoelectric performance of PVDF [20]; and melt blending with zinc stannate (ZnSnO_3_) let PVDF achieve the high crystallinity and β-phase content [21]. Good dispersion of the functional fillers in PVDF matrix is still a challenge, especially for micro-injection molding which requires more for the fluidity of the polymer melt, resulting in the scarcity of studies on micro-injection molding of PVDF-based nano-composites today.

By adopting the special equipment self-designed in our research group, i.e., pan-mill mechanochemical reactor, which could exert strong multiple stresses, including squeezing, shear and hoop stress on materials in-between, PVDF active powder and PVDF/functional filler composite powder, e.g., PVDF/graphene, with high content of β-phase crystals (more than 90%) were obtained in our laboratory, and the conventionally injected samples with good comprehensive properties, especially dielectric properties, were also prepared [22,23]. This provides a good base for the micro-injection of PVDF and its nano-composites.

Accordingly in this work, PVDF/graphene composite powder with extremely high content of β-phase crystals (>90%) obtained on our pan-mill mechanochemical reactor was adopted as raw material to be micro-injection molded on a Micro Power 5 micro-injection molding machine. The effects of the most important two factors, i.e., injection speed and cavity temperature, on the filling performances of this nano-composite during the extremely narrow channel of the micro-injection molding, as well as the reproducibility of the micro-injection molding parts were systematically studied. Via comparing with the conventional injection molding, the crystalline structure, especially the β-phase content and mechanical properties of the PVDF/graphene micro-injection parts were investigated. The results would provide a theoretical guidance for the preparation and application of the micro-electronic devices of the PVDF/graphene composite with high β-phase crystal content.

## 2. Experimental

### 2.1. Materials

Polyvinylidene fluoride (PVDF, FR906, ρ = 1.76 g/cm^3^, T_m_ = 173 °C) was purchased from Shanghai 3F New Materials Co., Ltd., China. Graphene (GP, SE1231) was provided by Sixth Element (Changzhou) Materials Technology Co., Ltd., Changzhou, China.

### 2.2. Sample Preparation

PVDF/GP composite powder was obtained by co-milling certain PVDF and GP on our self-designed pan-mill mechanochemical reactor [24] at room temperature. The milling cycle was 10 times. GP content in the co-milling powder was 1 wt%. The heat generated during the milling process was carried out by circulating cooling water.

The co-milling powder was then extruded and granulated on a parallel twin-screw extruder (TSSJ-25, Keqiang Chemical Equipment Company of China) to get PVDF/GP composite granules, which would be injected into dumbbell-type standard specimens (length × width × thickness: 150 mm × 10 mm × 4 mm) for a mechanical property test on a traditional injection molding machine (Ningbo Haitian Group Co., Ltd., Ningbo, China), marked as CIM. Before the test, the specimens were stored at room temperature for 48 h. The un-milled PVDF/GP composite was directly extruded, granulated and injection molded as the control sample.

PVDF and PVDF/GP micro-injection molding parts with the dimensions in Figure 1 were prepared by using the Micro Power 5 injection molding machine (Battenfeld Company, Vienna, Austria).

### 2.3. Characterization

Scanning electron microscope (SEM) was conducted to observe the morphologies of PVDF/GP powder and the fracture sections of the macro-/micro-parts by using an INSPECT F scanning Electron Microscope (FEI Company, Hillsboro, OR, USA) at the voltage of 20 kV. Before the test, the samples were sprayed with gold.

Ultra-depth three-dimensional microscope was performed to observe the three-dimensional morphology of the micro-protrusions on both sides of the micro-injection sample by using a VHX-10000 ultra-depth three-dimensional microscope (Keyence Company, Osaka, Japan).

The melting and crystallization behavior of the PVDF and PVDF/GP composite were studied using TA-Q20 differential scanning calorimeter (TA Company, Newcastle, DE, USA) from room temperature to 200 °C under nitrogen atmosphere. The heating or cooling rate was 10 °C/min. The melting enthalpy (Δ*H*_m_) was obtained from the melting peak area, and the crystallinity of PVDF was calculated according to the following calculation formula:(1)$$X_{C}=\frac{\Delta H_{m}}{\Delta H_{100\%}}\times 100\%$$
where, Δ*H*_100%_ is the melting enthalpy for complete crystallization of the polymer, Δ*H*_m_ is the melting enthalpy measured by DSC, and for PVDF it is Δ*H*_100%_ = 105 J/g [25].

The thermal stability of the PVDF and PVDF/GP composite were studied using TA-Q50 thermogravimetric analyzer (TA Company, Newcastle, DE, USA) from room temperature to 700 °C under nitrogen atmosphere. The heating rate was 10 °C/min.

The crystalline structure of PVDF was analyzed by using DX-1000 X-ray diffractometer (Dandongfangyuan Instrument Factory, Shanghai, China). The diffraction intensity of Cu Ka radiation was assessed at a voltage of 40 kV and 25 m. The scanning angle was from 5° to 50° and the scanning speed was 0.02 °/min.

Fourier transform infrared (FTIR) spectra were carried out using a Nicolet 6700 spectrophotometer (Thermo Scientific, Waltham, MA, USA) in transmission scanning mode from 400 cm^−1^ to 4000 cm^−1^. The testing sample was obtained by co-grinding certain powder and KBr, and then pressing them into tablet. The scanning number was 128 times.

Mechanical properties were tested using a 5567-testing machine (Instron, Boston, MA, USA) at room temperature according to GB/T 1040.1-2018. The stretching speed was 50 mm/min. Five samples were tested for each group to calculate the average value.

## 3. Results and Discussion

### 3.1. Filling Behaviors of PVDF/GP Composites during the Extremely Narrow Channel

For the different cavity size of micro-injection molding from conventional injection molding, the established flowing and filling theories based on the conventional injection are no longer suitable. So, the studies on the filling behaviors of PVDF/GP composites at different process parameters in the extremely narrow channel of the micro-injection molding become necessary.

Filling behaviors during micro-injection molding are influenced by several processing conditions including injection speed, injection pressure, cavity temperature, barrel temperature, holding pressure and holding time [26], among which, injection speed and cavity temperature seem to be more important. By adopting the online monitoring system of Micro Power 5, the changes of injection pressure and cavity pressure, which could reflect the filling behaviors of polymer melt with the injection time were obtained, as shown in Figure 2.

As shown, the micro-injection molding process presented four stages, i.e., channel filling, cavity filling, pressure holding and cooling, respectively corresponding to the sections of No. 1–4, 4–6, 6–7 and the later part in Figure 2a. At the stage of channel filling, the polymer melt flew into the narrowed and cooling channel, its viscosity gradually increased, along with the increase of the injection pressure. For no melt in the cavity, the cavity pressure was zero (No. 1–2 in Figure 2b). Once the melt entered into the cavity, ascribing to the continuously compacted melt, the injection pressure rose sharply to the maximum, so did the cavity pressure (No. 2–5 in in Figure 2b). However, with the proceeding of the filling, the melt in the cavity gradually cooled and solidified, the injection pressure in the cavity dropped and last held at 800 bar, and the cavity pressure decreased (No. 5–6 in Figure 2b). After complete cooling and solidification, the product was demolded, and the injection molding was finished.

By using the average mass of ten fully filled micro-injection samples as the reference standard, the feeding rates of PVDF/GP composites were calculated according to the below formula, and the results were listed in Figure 3c,f:(2)$$\mathrm{Filling}\;\mathrm{rate}=\frac{\mathrm{Average}\;\mathrm{quality}}{\mathrm{Standard}\;\mathrm{quality}}\times 100\%$$

With the increase of both injection speed and cavity temperature, the filling rate of the polymer melt significantly increased. At low injection speed (100 mm/s) or cavity temperature (40 °C), the incomplete filling occurred due to the relatively low shear viscosity of polymer melt. When injection speed increased to 300 mm/s or cavity temperature rose to 80 °C, the full filling was achieved due to the proper shear viscosity of the melt. However, with the further increase of the injection speed (500 mm/s) or cavity temperature (120 °C), the viscosity of the melt was so high that overflowing occurred and the flash formed. All indicated that both the injection speed and cavity temperature had greater impact on the filling behavior of PVDF/GP system, it was vital to choose an appropriate injection speed and mold temperature.

### 3.2. Reproducibility of PVDF/GP Micro-Injection Parts

According to the filling behaviors of PVDF/GP composites, the proper processing parameters were adopted, i.e., melt temperature of 180 °C, holding pressure of 800 bar, holding time of 2 s and measurement volume of 1 mm. Under these processing conditions, the good micro-injection molding of the PVDF/GP composite was successfully realized. As shown in Figure 4a,b, the thinnest part was only 0.3 mm and the mass was only 0.03 g. The good replication and dimensional stability of the micro-protrusion structure was also confirmed by SEM images (Figure 4c), and to deeply illustrate the reproducibility of the micro-injection molding, an ultra-depth three-dimensional microscope was further used, as shown in Figure 5. The different color represents different height. Obviously, almost all color bands distributed in a regular shape, indicating that PVDF/GP melt completely filled the cavity, and the surface of the micro-protrusion was smooth and defect-free.

### 3.3. Morphologies of PVDF/GP Micro-Injection Parts

A polarized light micrograph (PLM) photo of the ultra-thin section of the micro-injection part along the melt flow direction is shown in Figure 6. The micro-injection part presented the typical “skin-core” structure, which contained the cortex, the shear layer and the core layer. The cortex was solidified and formed once the polymer melts contacting the cold wall of the cavity during the process of the micro-injection molding. The cortex could act as the thermal insulation layer during the subsequent micro-injection, leading to the decrease in the cooling rate of the remained melt, and resulting in the formation of the oriented lamella structure at such high shear force caused by the extremely small runner and cavity of micro-injection molding, i.e., the shear layer. Under the combined influences of the cortex and the shear layer, the core layer had the relatively low shear rate and cooling rate, making the spherulite structure with high crystallinity formed in this layer. It should be noted that, during the conventional injection molding, for the relatively low injection speed, the shear rate from the center to the wall of the cavity changed little and no obvious “skin-core” structure could be formed.

The cortex and the shear layer of pure PVDF micro-parts at different injection speeds (100 mm/s, 300 mm/s and 500 mm/s) along the melt flow direction are shown in Figure 7. Clearly, with the increase of the injection speed, a more obvious fibrous structure appeared in the shear layer, while little change was found in the cortex, suggesting that only a high enough shear rate could induce the orientation of PVDF molecules. It was this strong shear action of micro-injection molding that reduced the viscosity of PVDF melt near the wall, disentangled the PVDF molecular chains and orchestrates the preferential arrangement and orientation along the flow direction. Moreover, the fast cooling rate in the micro-cavity, i.e., 0.1 s, further guaranteed the oriented PVDF molecules and so formed the fibrous structure in the shear layer. However, in the core layer, the shear force was too weak to induce the formation of the fibrous structure.

The morphologies of GP particles along the melt flow direction in the shear and core layers of PVDF/GP micro-/macro-parts could be compared from the SEM images of the fracture surfaces (Figure 8). The short white lines represent GP particles. The GP particles were uniformly dispersed in the PVDF matrix without obvious aggregations and phase interfaces, no matter whether in the micro- or macro-parts. In contrast, in the micro-part, some GP particles presented the significant orientation along the melt flow direction in the shear layer, but this orientation weakened in the core layer. While in the macro-part, no obvious orientation of the GP particle was found in both shear and core layers.

The formation of this special dispersion of GP particles in the micro-part could be attributed to its extremely high shear rate, i.e., 10^6^ s^−1^, almost 2 orders of magnitude higher than that of conventional injection molding. At such a high shear rate, GP agglomerates were suffering the high shear force and were liable to be exfoliated, orderly arranged, and even oriented along the flow direction of the melt. According to the theory of the injection molding, the shear rate from the center to the wall of the cavity presented a gradient distribution, i.e., the shear rate in the shear layer was greater than that in the core layer, resulting in a higher orientation degree of GP particles in the shear layer. In contrast, due to the relatively low injection speed of conventional injection molding and the large mold cavity, the shear force imposed on GP particles was too low to induce the orientation of GP particles along the melt flow direction.

### 3.4. Melt Behaviors of PVDF/GP Micro-Injection Parts

Figure 9a shows DSC heating curves of PVDF/GP micro-parts at different injection speeds, and the melting point T_m_ and crystallinity X_c_ are listed in Table 1. Significantly, T_m_ and X_c_ of the micro-parts are higher than that obtained from the conventional injection molding (CIM), and the higher the injection speed, the higher T_m_ and X_c_. For example, when the injection speed increased from 100 mm/s to 500 mm/s, T_m_ and X_c_ of the micro-part respectively increased from 170.3 °C to 173.6 °C and 54.0% to 59.1%. Compared with CIM, much higher shear force was generated during micro-injection molding for its narrow channel, forcing PVDF molecular chains to preferentially arrange along the melt flow direction, and induce the formation and growth of the crystal nucleus. Higher injection speed meant greater shear rate exerted to the melt, making PVDF molecular chains move more vigorously and be arranged more easily into the crystal lattice to form crystals. The extremely fast cooling speed (0.1 s) of micro-injection molding allowed the crystal structure to be maintained, further improving the crystallinity.

Fixing the injection speed, the effects of cavity temperature on the DSC heating curves of PVDF/GP micro-parts are shown in Figure 9b, and T_m_ and X_c_ obtained from the DSC curve are listed in Table 1. With the increase of the cavity temperature from 40 °C to 120 °C, both T_m_ and X_c_ of the micro-parts increased. As known, higher cavity temperature was beneficial to the movement and arrangement of PVDF molecular chains into the crystal lattice, thus, promoting the formation of a more perfect crystal and increasing the crystallinity of the sample. Strangely, at the cavity temperature of 120 °C, a small new melting peak with the peak temperature of 180.8 °C appeared, probably ascribing to the formation of the γ-phase PVDF with the TTTGTTTG’ conformation. Normally, PVDF has five crystalline phases, i.e., α, β, γ, δ and ε, among which β-phase with all-trans planar zigzag (TTT) conformation exhibits quite a high dipole moment, so endows PVDF with excellent electro-active properties. However, β-phase PVDF is unstable and can only be obtained under special conditions, e.g., melt at high pressure, mechanical stretching, etc. [27]. Compared with β-phase, γ-phase presents weaker polarity, but still has strong electrical properties and a higher melting point, giving its extremely high added value. γ-phase PVDF could be formed via high temperature crystallization, the addition of the nucleating agent or in the restricted space. It was possible that under the extremely high shear force of micro-injection molding, β-phase formed first, then some folded chain lamellae formed on the surface of β-phase crystals. These lamellae would change into α-phase or γ-phase according to the temperature [28]. High cavity temperature, e.g., 120 °C, meant high crystallization temperature and low melt cooling rate, providing a long time for PVDF to crystallize, thus, promoting the formation of the γ-phase with higher melting temperature [29].

### 3.5. Crystal Phases of PVDF/GP Micro-Parts

The crystal phases of the PVDF/GP composite formed during micro-injection molding were systematically studied by XRD and FTIR tests. As shown in Figure 10a, α-phase PVDF presents three diffraction peaks, respectively located at 2θ = 18.30°, 19.90° and 26.56°, corresponding to (020), (110) and (021) planes. β-phase PVDF only has one overlapping characteristic peak at 2θ = 20.26°, i.e., (110) and (200) planes [30]. Obviously, micro-injection molding promoted the transformation of α-phase to β-phase, making the stronger β-phase peak at 2θ = 20.26° and weaker α-phase peak at 2θ = 26.56° appeared in the XRD curves of micro-parts, compared with CIM. With the increase of the injection speed, the α-phase peak at 2θ = 26.56° weakened further, and even disappeared when injection speed reached 500 m/s.

The FTIR test presented the same changes (Figure 10b), i.e., the characteristic vibration bands of α-phase at 766 cm^−1^ and 976 cm^−1^ weakened, while the β-phase peaks at 840 cm^−1^ and 1279 cm^−1^ enhanced. All indicated the increased content of β-phase PVDF in micro-part, attributing to the high shear force produced by the extremely high micro-injection speed on PVDF molecular chains, which made PVDF molecular chains untwisted and stretched along the flow direction, and then provided enough energy for α-phase with spiral TGTG’ conformation to transform into β-phase with zigzag TTTT all-trans conformation [31]. Moreover, the fast cooling speed of micro-injection also retarded the relaxation of the moving PVDF molecular segments, so could retain the generated β-phase.

According to FTIR results, and by using the following equation [30], the relative content of β-phase (*F*(*β*)) in PVDF was calculated, as shown in Figure 11:
(3)$$F(\beta )=\frac{X_{\beta }}{X_{\alpha }{+X}_{\beta }}=\frac{A_{\beta }}{(K_{\beta }/K_{\alpha })A_{\alpha }+A_{\beta }}$$
where X_*α*_ and X_*β*_ are the crystallinity of α-phase and β-phase; *A_α_* and *A_β_* represent the adsorbance at 766 and 840 cm^−1^; K_*α*_ and K_*β*_ are the adsorption coefficient, K_*α*_ = 6.1 × 10^4^ cm^2^ mol^−1^, K_*β*_ = 7.7 × 10^4^ cm^2^ mol^−1^.

The higher the injection speed, the more the β-phase in PVDF. In the macro-part (CIM), the β-phase content was only 42%; while in the micro-part, at the injection speed of 500 mm/s, the β-phase reached 78%, almost two times of the macro-part. In contrast to micro-injection molding, the conventional injection can only provide low injection speed, e.g., 35 mm/s, so its shear force was too weak to induce the transformation of α-phase to β-phase. The cooling rate in the conventional injection was so slow that even some oriented PVDF molecular chains would relax and recover to α-phase with the lowest energy, resulting in the failure to obtain high β-phase content. While for micro-injection molding, its strong shear force destroyed and pulled the original PVDF spherulites, making PVDF microlites rearranged along the flow direction of the melt, thus, promoting the transformation of α-phase to β-phase. Inevitably, the relax of PVDF molecular chains happened during the micro-injection molding, but shear force should be absolutely dominant, well-maintaining the β-phase crystals induced by stress [32], thus, endowing the micro-part with high β-phase content.

However, the cavity temperature, which decides the cooling rate, has contrary influence on the crystal phases of PVDF/GP micro-parts. With the increase of the cavity temperature, the cooling rate of the melt slowed down, the thermal movement of PVDF molecular chains became intense, so PVDF molecular chains were more easily disoriented to form the most stable α-phase, making the α-phase peak at 2θ = 26.56° gradually enhanced and the β-phase peak at 2θ = 120.26° weakened (Figure 12a). This was similar to the FTIR characteristic peaks of α-phase at 976 and 776 cm^−1^ as well as β-phase at 840 and 1279 cm^−1^ (Figure 12b). Strangely, at the cavity temperature of 120 °C, the β-phase at 840 cm^−1^ broadened and seems to contain a shoulder peak at 833 cm^−1^. Meanwhile, another shoulder peak appeared at 1234 cm^−1^. Both belonged to the γ-phase. The formation of these γ-phase peaks could be ascribed to the low cooling rate, high crystallization temperature and long crystallization time at 120 °C. [33,34].

### 3.6. Mechanical Properties of PVDF/GP Micro-Parts

The transformation of the crystal phases has a profound impact on the mechanical performance of PVDF-based materials. Compared with the macro-part, the micro-part presented a different fracture behavior, with higher tensile strength and increased elongation at break (Figure 13). Its tensile strength was 57.6 MPa, ~7 MPa higher than that of the macro-part, attributing to the special orientation structure and higher crystallinity. Its elongation at break reached 53.6%, ~227% higher that of the macro-part. This could be ascribed to the greatly enhanced β-phase with high polarity and fibrous shape in the micro-part (high to 78%, almost 2 times of that of the macro-part), which endowed the micro-part with more efficiency energy dissipation under the stress field [35], so greatly increased the toughness of the micro-part.

## 4. Conclusions

The filling behaviors and the structure evolution of the PVDF/GP composites during the extremely narrow channel of micro-injection molding were systematically studied. Under proper parameters, the complete filling of the composite as well as the micro-part with good replication and dimensional stability were achieved. Compared with conventional injection molding, the extremely high injection speed and small cavity brought high shear force and cooling rate for the composite melt, leading the obvious “skin-core” structure of the micro-parts and the orientation of both PVDF and GP in the shear layer, which was not observed in the macro-parts. The different structure of the micro-parts from the macro-parts endowed them with a higher melting point and crystallinity, which was further increased with the increase of the injection speed and cavity temperature. The high shear force field of micro-injection could induce the transformation of α-phase PVDF to β-phase, and the higher the injection speed, the more the β-phase content. At the injection speed of 500 mm/s, the β-phase content reached 78%, almost 2 times that of the macro-part. However, with the increase of cavity temperature, the unstable β-phase gradually transferred to α-phase, even γ-phase, for the enhanced movability of PVDF molecular chains. The micro-part had the higher tensile strength (57.6 MPa) and elongation at break (53.6%) than those of the macro-part due to its increased crystallinity and β-phase content.

## Figures and Tables

**Figure 1 polymers-13-03292-f001:**
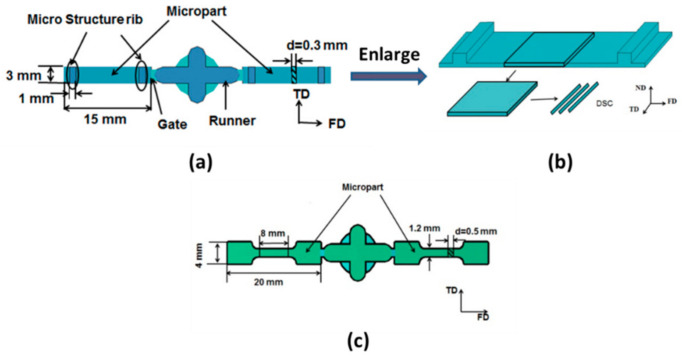
Dimensions of the micro-part (**a**,**b**) and the dumbbell-shaped micro-part (**c**).

**Figure 2 polymers-13-03292-f002:**
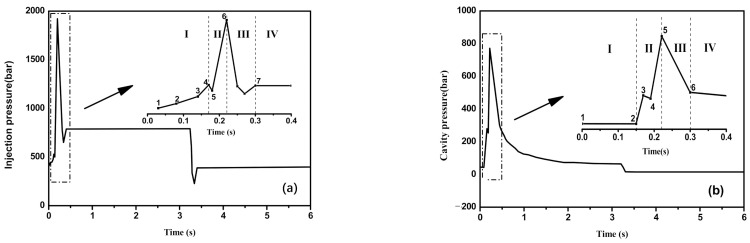
Injection pressure (**a**) and cavity pressure (**b**) vs. injection time of PVDF/GP composite with 1 wt% GP during the micro-injection molding process.

**Figure 3 polymers-13-03292-f003:**
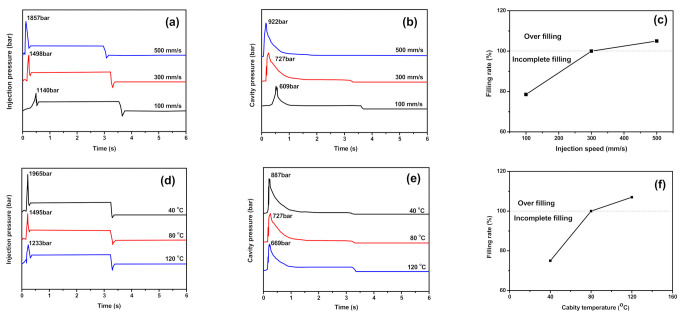
Injection pressure and cavity pressure vs. time of PVDF/GP composites with 1 wt% GP during the micro-injection molding process at different injection speed (**a**–**c**) and cavity temperature (**d**–**f**).

**Figure 4 polymers-13-03292-f004:**
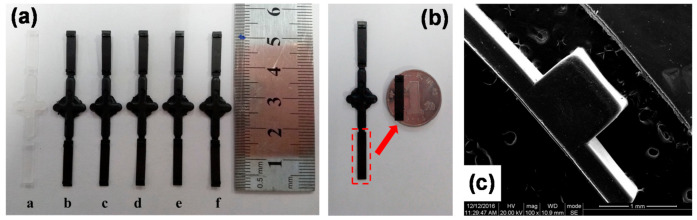
Photos (**a**,**b**) and SEM images (**c**) of PVDF/GP micro-parts with 1 wt% GP.

**Figure 5 polymers-13-03292-f005:**
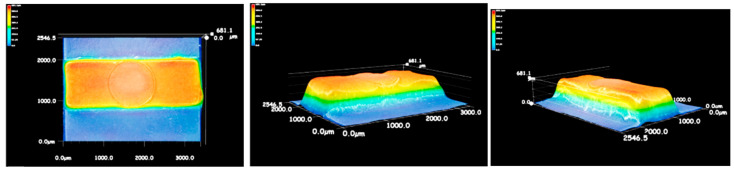
3D images of the mold structure of PVDF/GP micro-part (GP content: 1 wt%).

**Figure 6 polymers-13-03292-f006:**
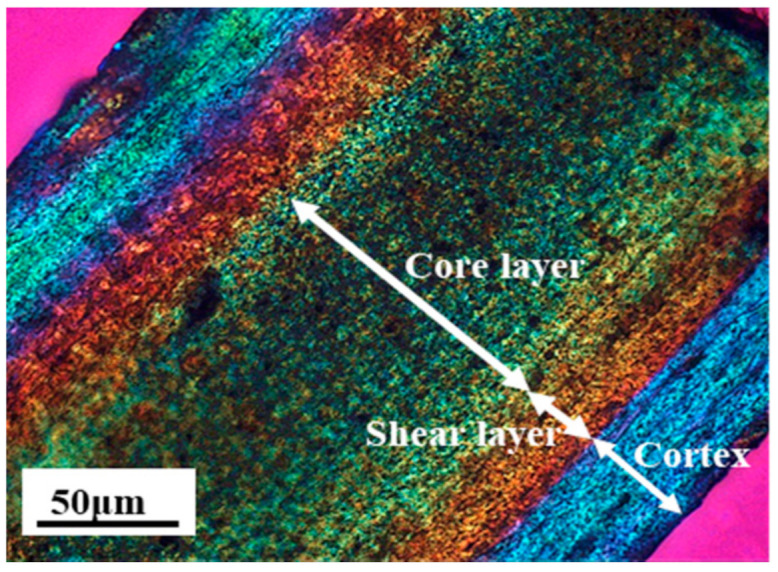
Polarized light micrograph of PVDF/GP micro-part (GP content: 1 wt%).

**Figure 7 polymers-13-03292-f007:**
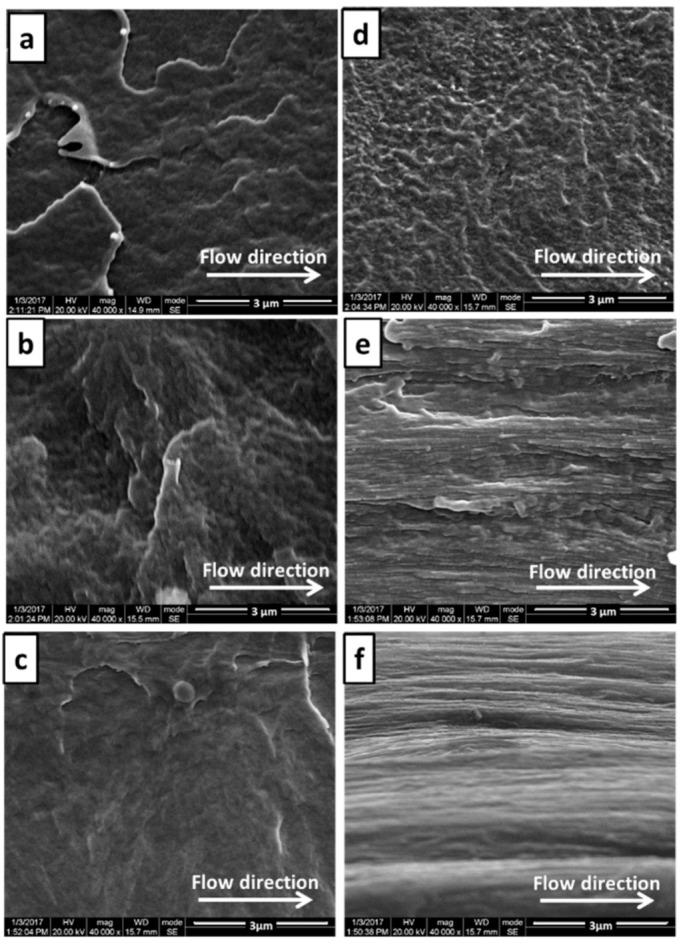
SEM images of the core layers and the shear layers of PVDF micro-parts at different injection speed: the core layer at 100 mm/s (**a**), 300 mm/s (**c**) and 500 mm/s (**e**); the shear layer at 100 mm/s (**b**), 300 mm/s (**d**) and 500 mm/s (**f**).

**Figure 8 polymers-13-03292-f008:**
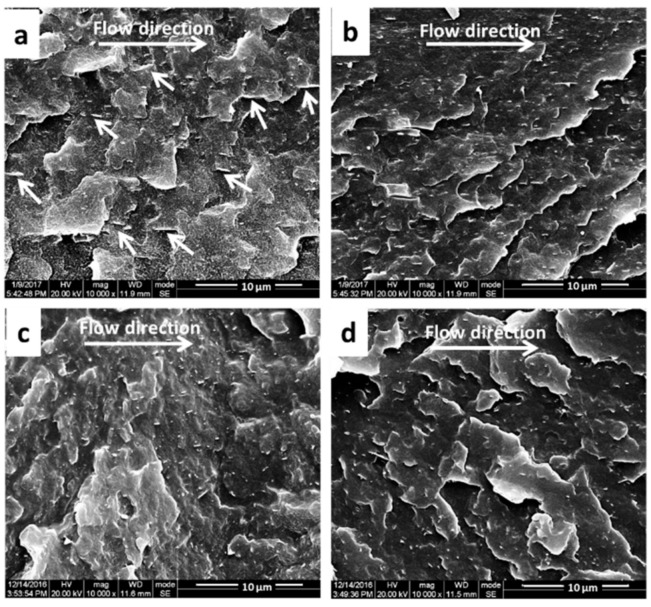
SEM images of the fractured surfaces of the shear layer (**a**) and the core layer (**b**) of PVDF/GP micro-part; the shear layer (**c**) and core layer (**d**) of PVDF/GP macro-part. (GP content: 1 wt%).

**Figure 9 polymers-13-03292-f009:**
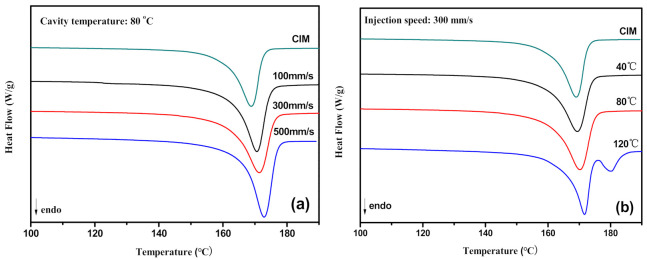
DSC curves of PVDF/GP micro-parts at different injection speeds (**a**) and different cavity temperature (**b**) (GP content: 1 wt%).

**Figure 10 polymers-13-03292-f010:**
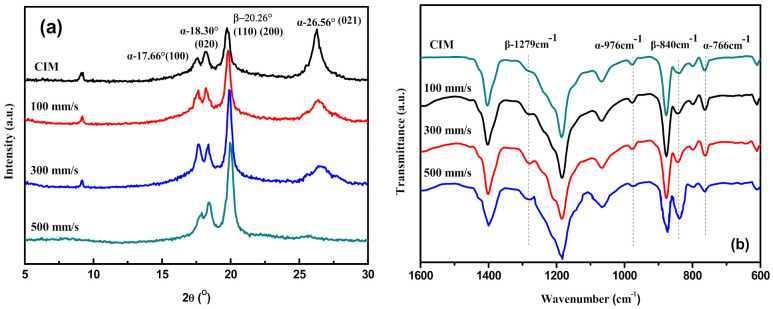
XRD spectra (**a**) and FTIR spectra (**b**) of PVDF/GP micro-parts at different injection speeds (Cavity temperature: 80 °C; GP content: 1 wt%).

**Figure 11 polymers-13-03292-f011:**
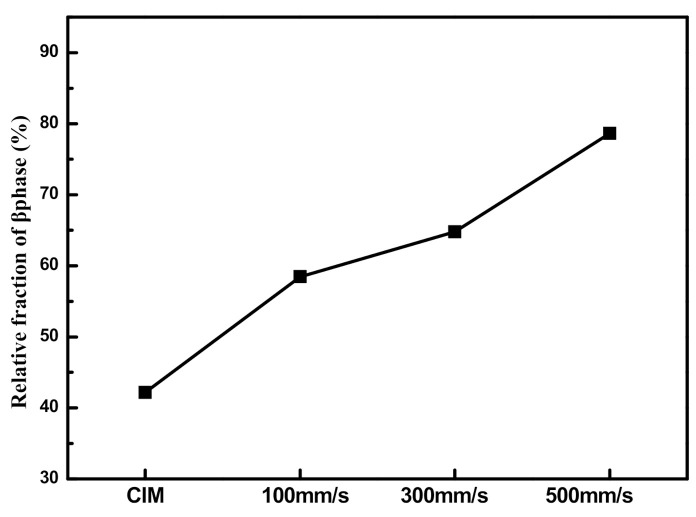
Relative fraction of β-phase PVDF in PVDF/GP micro-parts at different injection speeds (Cavity temperature: 80 °C; GP content: 1 wt%).

**Figure 12 polymers-13-03292-f012:**
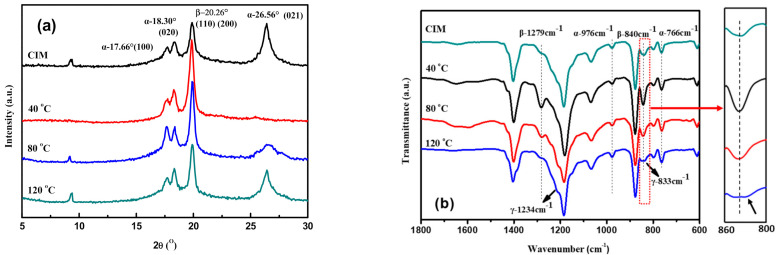
XRD spectra (**a**) and FTIR spectra (**b**) of PVDF/GP micro-parts at different cavity temperature (Injection speed: 300 mm/s; GP content: 1 wt%).

**Figure 13 polymers-13-03292-f013:**
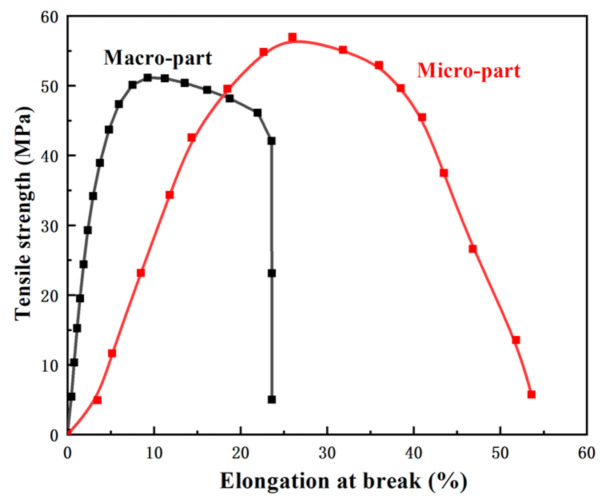
Stress-strain curves of PVDF/GP macro- and micro-parts (GP content: 1 wt%).

**Table 1 polymers-13-03292-t001:** Thermal analyses of PVDF/GP micro-parts at different injection speeds and different cavity temperature.

		T_m_/°C	ΔH_m_/J·g^−1^	X_c_
CIM		169.6	55.6	52.9
Injection speed (mm/s)	100	170.3	56.7	54.0
	300	171.6	58.6	55.8
	500	173.6	62.6	59.1
Cavity temperature °C	40	170.5	57.5	54.8
	80	171.6	59.6	56.7
	120	172.2/180.8	60.1	57.2

## Data Availability

Not applicable.

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
