# Peer review of "Fabrication of Poly(Vinylidene Fluoride)/Graphene Nano-Composite Micro-Parts with Increased β-Phase and Enhanced Toughness via Micro-Injection Molding"

_polymers, 2021, doi:10.3390/polym13193292_

Round 1

Reviewer 1 Report

The paper is alltogether round but needs some improvement. Regarding English: you could make shorter sentences to make it easier for the reader.

Abstract:

line 9: 'shear layer'

Introduction

p. 2:

line 3: 'limited to materials'  split this sentence.

line 7/8: 'functionality'

line 18: give the full name for abbreviations when they occur first (also for 'PLM' on p. 6)

line 22: 'achieve' - split this sentence

last paragraph: 'Accordingly in this work;'

chapter 3.1

p. 4: last paragraph: hy should the viscosity increase in narrow channels? Ususally polymers show a shear-thinning behaviour.

p. 5: last paragraph: split the sentence 'At low injection speed ...'

chapter 3.2

p. 6:

line 7: 'confirmed'

Please check the layers shown and named in Fig. 6. The blue layer seems to be the layer on the surface of the part and not a core layer. Rethink the place in the part and the naming of the layers and check your discussions involving the layers regarding this.

chapter 3.3

p. 8: line 12: 'suffering'

chapter 3.4

p. 8: line 4 from the bottom: 'arrranging'

p. 8: line 2 from the bottom: 'move'

chapter 3.5

p. 12 top: you mention a shoulder peak at 833 cm-1: this is not really visible. If there is one please add a zoom of thgis section

Author Response

Point 1: p. 4: last paragraph: why should the viscosity increase in narrow channels? Ususally polymers show a shear-thinning behaviour.

Response 1: Thanks for the reviewer’s question. Yes, polymers usually show a shear-thinning behaviour. But in my opinion, it is just for the stable flow process. Here, we focused on the filling behaviour of polymer melt into the narrow channel, i.e. from entering to complete filling the channel. During this process, the polymer melt continually suffered the resistances from the narrow channel and the melt that had entered the channel, leading to the gradually increase of its viscosity.

Point 2: Please check the layers shown and named in Fig. 6. The blue layer seems  to be the layer on the surface of the part and not a core layer. Rethink the place in the part and the naming of the layers and check your discussions involving the layers regarding this.

Response 2: Very sorry for our carelessness. We have checked the layers in Fig. 6,and corrected the marks in the figure. We have also checked the discussions involving the layers, they are OK.

Point 3: p. 12 top: you mention a shoulder peak at 833 cm-1: this is not really visible. If there is one please add a zoom of this section.

Response 3: Thanks for the reviewer’s suggestion, we have add a zoom of this section.

Reviewer 2 Report

The author fabricated micro-parts made of poly(vinylidene fluoride)/graphene (PVDF/GP) composite successfully with good replication and dimensional stability via micro-injection molding. The filling behaviors and the structure evolution of the composite during the fabrication process were systematically studied. The results showed that the β-phase PVDF in micro-part was 78%, which was almost 2 times of that in macro-part. Moreover, the micro-part had similar tensile strength to the macro-part, but higher elongation at break (67.2%), about three times as much as that of the macro-part, confirming the improvement of the toughness for the increased β-phase content. The work is interesting and can be published in Polymers if the following issues can be addressed: 1. The authors should cite the paper “Nanocomposites for electronic applications that can be embedded for textiles and wearables” (Science China Technological Sciences, 2019, 62, 895-902) and “Strong, lightweight, and highly conductive CNT/Au/Cu wires from sputtering and electroplating methods” (Journal of Materials Science & Technology, 2020, 40, 99-106) in the introduction section for better review about the fabrication and application of polymer based nano-micro composite. 2. Why were GP contents from 0.5 wt% to 2.5 wt% mentioned in section 2.2 but only data of 1 wt% GP were presented in the manuscript? 3. What standard was applied for the tensile test? Why were the stretching speed of 50 mm/min and 5mm/min chosen? 4. How the elongation at break was determined? The authors should consider the measurement accuracy if it was based on the crosshead movement. 5. Larger scale bars are required for Figures 7 and 8 6. In Figure 13, strain has no unit. The author should choose either elongation (%) or strain to show in the Figures. The “1% GP” should be use in the legend 7. From the stress-strain curves shown in Figure 13, strength and Young’s modulus of the microparts were lower than those of the macropart. What could be the reasons? 8. The authors mentioned that “When considering the effects of the reinforcing fillers on elongation, a decrease ra-ther than an increase is usually expected.” That is only true when comparing with neat polymer part without fillers. However, this study compared two parts made of the same composite but by different fabrication methods. The authors should clarify this in the revised manuscript. 9. There are several spelling errors (ex: tansformation, cabity…) in the manuscript. The authors should correct them.

Author Response

Point 1: The authors should cite the paper “Nanocomposites for electronic applications that can be embedded for textiles and wearables” (Science China Technological Sciences, 2019, 62, 895-902) and “Strong, lightweight, and highly conductive CNT/Au/Cu wires from sputtering and electroplating methods” (Journal of Materials Science & Technology, 2020, 40, 99-106) in the introduction section for better review about the fabrication and application of polymer based nano-micro composite.

Response 1: Thanks for the reviewer’s suggestion, we have cited these two papers in the introduction section.

Point 2: Why were GP contents from 0.5 wt% to 2.5 wt% mentioned in section 2.2 but only data of 1 wt% GP were presented in the manuscript?

Response 2:  Very sorry for our carelessness. We indeed studied the co-milling composites with several GP contents, i.e. 0.5 wt%, 1 wt%, 1.5 wt%, 2 wt% and 2.5 wt%. But after considering the comprehensive properties including mechanical properties, dielectric properties and thermal conductivity of the composites, we chose the composite with 1 wt% GP to study its micro-injection process. Accordingly, we have revised the description in section 2.2.

Point 3: What standard was applied for the tensile test? Why were the stretching speed of 50 mm/min and 5mm/min chosen?

Response 3: Thanks very much for the reviewer’s questions. We noted that the stretching speeds for macro- and micro-parts were different. To better compare the mechanical properties of these two parts, we have repeated the mechanical test of the micro-part by using the stretching speed of 50 mm/min. According to the test results, we have revised the relative discussion. The standard for the tensile test was GB/T 1040.1-2018, which has been added in the 2.3 section.

Point 4: How the elongation at break was determined? The authors should consider the measurement accuracy if it was based on the crosshead movement.

Response 4: For macro-part, the elongation at break was tested by the strain sensor of the 5567 Instron. While for micro-part, the elongation at break was tested according to the distance between two fixtures. To ensure the measurement accuracy, the position of the fixture was set near the center part.

Point 5: Larger scale bars are required for Figures 7 and 8

Response 5: We have enlarged the scale bars in Figures 7 and 8.

Point 6: In Figure 13, strain has no unit. The author should choose either elongation (%) or strain to show in the Figures. The “1% GP” should be use in the legend

Response 6: We have revised Figure 13.

Point 7: From the stress-strain curves shown in Figure 13, strength and Young’s modulus of the microparts were lower than those of the macropart. What could be the reasons?

Response 7: We noted that the stretching speeds for macro- and micro-parts were different. To better compare the mechanical properties of these two parts, we have repeated the mechanical test of the micro-part by using the stretching speed of 50 mm/min. At this same stretching speed, micro-part presents higher tensile strength and elongation at break, ascribing to its higher crystallinity and the increased β-phase. We have revised the discussions in the 3.6 section.

Point 8: The authors mentioned that “When considering the effects of the reinforcing fillers on elongation, a decrease rather than an increase is usually expected.” That is only true when comparing with neat polymer part without fillers. However, this study compared two parts made of the same composite but by different fabrication methods. The authors should clarify this in the revised manuscript

Response 8: Thanks very much for the reviewer’s comments. Yes, that is only true when comparing with neat polymer without fillers. So, we have deleted this mention.

Reviewer 3 Report

The journal Polymers is a suitable one for the present manuscript which has an intersting topics in polymers field regarding fabrication of Poly(vinylidene fluoride)/Graphene Nano-composite Micro-parts with Increased β-Phase and Enhanced Toughness via Micro-injection Molding. Such   things are strong points but before publications the authors have to complete and correct the week points  according to following comments:

the introduction needs to be completed with more details regarding the paper topics novelty in recent literature due to the fact that the majority of refernces are older than 3 years

2 the paper is clearly written but is not well organized having only one table and 13 figures : I do recommend that a part of the figures data (such as example XRD and FT-IR spectra) to be put it in the table

3 to introduce standard deviation in the values of table 1

4 in FiG 10 a there are XRD spectra and the term “ curves”  is not properly used

5 in the title the word via has to be written via with italic

6 some letters are written with small letter instead of capitals ( see reference 21

Author Response

Point 1: the introduction needs to be completed with more details regarding the paper topics novelty in recent literature due to the fact that the majority of refernces are older than 3 years

Response 1: Thanks very much for the reviewer’s suggestion. Accordingly, we have added more details regarding the innovation of this manuscript. Compared with conventional injection molding, micro-injection molding puts forward quite high requirements for the rheological properties of a polymer material, making it only suitable for some single thermoplastics. In this manuscript, we prepared PVDF/GP composites with well dispersion of GP particles and high β-phase crystals, realized their micro-injection molding and systematically studied the filling behavior and the structure evaluation of the composites during the micro-injection process. This is our innovation, and would provide a theoretical guidance for the preparation and application of the micro-electronic de-vices of PVDF based composites.

Point 2: the paper is clearly written but is not well organized having only one table and 13 figures: I do recommend that a part of the figures data (such as example XRD and FT-IR spectra) to be put it in the table

Response 2: Thanks very much for the reviewer’s suggestions. However, XRD and FT-IR spectra were tested to directly see the changes of the crystal phases of PVDF, including the peak position, peak strength, and even the appearance of the new peak. If the figures date were put in the table, no intuitive observation effect was obtained. So, after the comprehensive consideration, we didn’t put a part of the figures date in the table.

Point 3:  to introduce standard deviation in the values of table 1

Response 3: Thanks for the reviewer’s suggestion. However, ascribing to the high resolution of DSC test, it is seldom to do DSC test of one sample for more than 2 times, so usually, no standard deviation is required for the data from DSC curves. This is quite different from the mechanical test, in which at least five times for one sample are needed, so the standard deviation is required to be provided.

Point 4: in FiG 10 a there are XRD spectra and the term “ curves”  is not properly used, in the title the word via has to be written via with italic, some letters are written with small letter instead of capitals ( see reference 21)

Response 4: Very sorry for our carelessness. We have checked and corrected the mistakes.
